# Genome-Wide Association Studies of Key Traits in *Apis cerana cerana* (Hymenoptera: Apidae) from Guizhou Province

**DOI:** 10.3390/genes16101148

**Published:** 2025-09-27

**Authors:** Yinchen Wang, Changshi Ren, Yang Yuan, Xu Yang, Mengqing Deng, Tian Zhao, Rongqing Ren, Yan Liao, Hua Wang, Ziwei Jiang, Xiaofeng Xue, Xiaoming Fang

**Affiliations:** 1Guizhou Institute of Animal Husbandry and Veterinary Science, Guiyang 550002, China; changshia2022@163.com (C.R.); yy17783138675@163.com (Y.Y.); zixiyang147@163.com (X.Y.); xmsdmq@163.com (M.D.); zhaotianyeah@163.com (T.Z.); mrren2023@163.com (R.R.); m13981797280@163.com (Y.L.); wanghua7277@126.com (H.W.); 15185775162@163.com (Z.J.); 2Apicultural Research Institute, Chinese Academy of Agricultural Sciences, Beijing 100093, China; xue_xiaofeng@126.com (X.X.); fangxiaoming@caas.cn (X.F.)

**Keywords:** bee, Chinese honeybee, *Apis cerana cerana*, genome-wide association study, genes, morphological traits

## Abstract

**Background/Objectives**: This study aimed to identify genes linked to phenotypic traits in *Apis cerana cerana* through a genome-wide association study. **Methods**: Genomic data was collected from 116 workers across 12 regions in Guizhou Province, China, and 15 morphological traits were measured, including proboscis length, femur length, tibia length, tarsus length, tarsus width, tergite III and IV length, sternite III length, wax mirror length on sternite III, wax mirror slanted length on sternite III, wax mirror interval on sternite III, sternite VI length, sternite VI width, forewing length, forewing width, and cubital index. Then, a genome-wide association study was performed on these traits. **Results**: The analysis identified 12 SNPs significantly associated with tergite III and IV length, along with 2 SNPs linked to wax mirror length on sternite III, and 7 SNPs related to the wax mirror interval on sternite III. Eleven candidate genes for tergite III and IV length, two genes for wax mirror length on sternite III, and seven genes for wax mirror interval on sternite III were identified. These genes encode proteins involved in Longitudinals, Zinc, Lamin, BTB/POZ, Dyneins, and Phospholipases. **Conclusions**: The discovered SNPs and their corresponding genes may regulate the lateral and longitudinal development of the tergum and sternum in the *A. c. cerana*. Continued in-depth research on these aspects will help clarify how these SNPs regulate the tergum and sternum, thereby enhancing economic returns for beekeepers and promoting the conservation of germplasm resources in the native *Apis cerana cerana*.

## 1. Introduction

Honeybees are recognized as resource insects due to their indispensable role in maintaining ecosystem balance and promoting sustainable development [1]. *Apis cerana cerana* is an endemic species widely distributed across China. Guizhou Province, located in Southwestern China, harbors rich ecological resources characterized by high vegetation diversity, significant altitudinal variation, and a dynamic climate often described as “Weather varies within 10-*li* (5 km).” Geographic isolation and high biodiversity have led to pronounced genetic differentiation in *A. cerana cerana*, resulting in adaptive divergence to distinct habitats [2,3]. Such differentiation may be genetically regulated and reflected in the species’ morphological traits [4].

Single-nucleotide polymorphisms (SNPs) refer to variations in DNA sequences caused by single-nucleotide substitutions, transversions, insertions, or deletions [5]. As the most abundant and heritable genetic variations in genomes, SNPs play a pivotal role in phenotypic diversity and adaptive evolution. Compared to traditional molecular markers (e.g., microsatellites), SNPs offer advantages such as genome-wide distribution, high stability, and suitability for high-throughput detection [6]. Due to their abundance and uniform genomic distribution, SNPs are widely used in genome-wide association studies (GWAS) for analyzing genetic diversity and economically important traits in plants and livestock [7,8,9,10,11,12]. They also provide new insights into identifying candidate genes regulating bone growth, muscle development, and growth-related traits, as well as marker-assisted breeding.

Previous research shows that SNPs are linked to immune response, behavioral traits, and economically significant characteristics in *Apis mellifera* [13,14]. For instance, Zayed et al. found that coding SNPs in *A. mellifera* exhibit higher *Fst* estimates than non-coding SNPs, suggesting positive selection-driven adaptive evolution. They also observed a negative correlation between Fst and local GC content around coding SNPs, indicating that AT-rich genes play a crucial role in adaptive evolution [15]. Spötter et al. validated SNP markers associated with mite resistance using an *A. mellifera* SNP chip [16]. Similarly, Bessoltane et al. identified recombination signals in honeybee chromosomes using SNP-based analysis [17]. Parejo et al. demonstrated that only 50 SNPs are sufficient to distinguish native from introduced honeybee populations, highlighting the utility of low-density SNP panels for population management and genetic conservation [18]. In *A. cerana cerana*, Hassanyar et al. discovered two SNPs significantly associated with resistance to Sacbrood virus [19].

Genome-wide association studies (GWAS) leverage natural populations to integrate genome-wide SNP data and phenotypic information, enabling efficient mapping of genetic loci underlying complex traits [20,21]. By screening high-density genetic markers across the genome, GWAS identifies SNPs significantly associated with phenotypes [22,23], offering novel strategies for uncovering candidate genes related to growth, development, and economically important traits, as well as facilitating marker-assisted selection. As a rapidly emerging genomic analysis tool, GWAS has revolutionized genomic research and is widely applied in humans, plants, animals, and microbes to dissect complex traits. In *A. mellifera*, GWAS has successfully identified SNPs linked to mite resistance and reproductive traits [16].

With the release of the *A. cerana cerana* genome assembly [24,25,26], whole-genome resequencing now enables the detection of extensive genetic variations in individuals or populations. This approach enhances the resolution and power of traditional genetic methods and facilitates the identification of genomic regions under selection for local adaptation and economically important traits [26,27]. Combining SNP-based GWAS with phenotypic data can further elucidate the genetic basis of foraging and flight-related traits. For example, some authors used GWAS to reveal co-evolutionary relationships between floral genes and pollinator behavior in bees and hummingbirds [28,29]. Guichard et al. identified quantitative trait loci (QTLs) associated with gentleness in *A. cerana cerana* and *A. mellifera* through GWAS [14]. He [30] identified regulatory genes controlling major traits in *A. cerana cerana* across 19 geographic populations using GWAS.

In this study, we performed whole-genome resequencing on 116 *A. cerana cerana* workers from diverse regions in Guizhou Province and conducted GWAS on 15 key morphological traits. Our objectives were to (1) identify functional SNPs significantly associated with these traits; (2) mine candidate genes with strong trait correlations; and (3) provide insights into the genetic evolution and conservation of local *A. cerana cerana* populations.

## 2. Materials and Methods

### 2.1. Source of Experimental A. cerana cerana Workers

The *A. cerana cerana* samples in this study were collected from 12 different representative locations in Guizhou Province, totaling 116 adult workers. The sampled bees were divided into 12 groups according to geographical names, with 6 replicates for the Xueshan group and 10 replicate samples for each of the remaining regions. The collection locations, altitudes, and breeding methods are shown in Table 1. The sampling targets were healthy colonies with normal population strength, no introduced subspecies, and originating from wild natural swarms. The sampling period was from June to July 2022, which is the season with a favorable climate, environment, and nectar/pollen sources in Guizhou. All bee samples were stored at −80 °C for further DNA extraction and sequencing. The sampled apiaries had no history of improvement or selective breeding, making them representative sampling sites of local genetic resources.

### 2.2. Phenotypic Measurement of A. cerana cerana

This experiment was conducted in accordance with the standard morphological indicator protocol and color criteria for honeybees [31,32]. Full-body images were captured, including the color patterns of the specimens. The forewings and hindwings were then carefully detached from the thorax using forceps. The remaining sternal parts were isolated by carefully separating the connective tissues between the wings. The dissected sterna were gently cleaned with a soft brush to remove any residual tissue and mounted on experimental boards using transparent tape. A specialized measuring microscope and its associated software (ImageView 3.7) were used to measure 15 morphological parameters after scale calibration: proboscis length, femur length, tibia length, tarsus length, tarsus width, tergite III and IV length, sternite III length, wax mirror length on sternite III, slanted wax mirror length on sternite III, wax mirror spacing on sternite III, sternite VI length, sternite VI width, forewing length, forewing width, and cubital index. Descriptive statistics were computed from the phenotypic measurements after the removal of outliers. Normality of the data was assessed using R 4.3.2. Only normally distributed data were retained for subsequent genome-wide association study (GWAS) analysis.

### 2.3. Genomic DNA Extraction and Resequencing Library Construction

Workers from 12 locations were selected, and total genomic DNA was extracted from their thoracic tissues using a Universal Genomic DNA Kit (Cat.No.CW0553, ComWin Biotech, Beijing, China) following the manufacturer’s protocol. The obtained genomic DNA was dissolved in TE buffer, examined for degradation and impurities using 1% agarose gel electrophoresis, and assessed for purity and concentration using a NanoDrop One spectrophotometer (Thermo Scientific, Waltham, MA, USA). Approximately 1 μg of genomic DNA template was used for resequencing library preparation following the TruSeq DNA Sample Preparation Guide (15026486 Rev.C, Illumina, San Diego, CA, USA). Specifically, DNA was randomly fragmented by ultrasonication, 350 bp fragments were selected for end repair and 3′-end A-tailing reactions, sequencing adapters were ligated, and target fragments were enriched by PCR to form sequencing libraries. Qualified libraries were subjected to paired-end sequencing (PE150) on the HiSeq X Ten platform at Novogene Co., Ltd. (Beijing, China), generating 150 bp raw reads. Raw reads were filtered to remove adapter sequences. For detailed methodology, please refer to our previous publication [33].

### 2.4. Sequence Alignment and SNP Detection

FastQC (v1.11.4) was used to verify raw read quality, and cutadapt removed adapter sequences, primers, poly-A tails and low-quality reads. Clean reads were aligned to the reference *A. cerana* genome (*Apis_cerana*.ACSNU2.0) using BWA (v0.7.17). Sequencing depth and genome coverage information for each sample were collected for variant detection. SNP variants were detected using GATK’s HaplotypeCaller (v4.2.0), then annotated with SnpEff 5.1.

### 2.5. Analysis of Kinship Among Individuals

Principal component analysis (PCA) was performed on filtered SNP data using GCTA based on genomic SNP differences, generating eigenval and eigenvec files. PCA plots were created in R 4.3.2 with PC1, PC2 and PC3 as *x*-axis, *y*-axis and *z*-axis. GCTA was used to construct genomic G matrices to better reflect kinship [34,35]. Using the PopLDdecay software (V3.43), we analyzed linkage disequilibrium (LD) decay in quality-controlled SNPs [36], comparing LD decay (to 0.1) distances and average marker distances to assess genome-wide coverage adequacy. Population genetic structure and lineage information were constructed using ADMIXTURE software (V1.3).

### 2.6. Genome-Wide Association Study

Quality-controlled SNPs were analyzed for 15 traits in 116 *A. cerana cerana* workers using GEMMA (v0.93) mixed linear models accounting for population stratification:Y = Wa + xB + u + E,u~MVN, (0, AT1K), ~MVNn (0, T1) where Y is the n × 1 trait vector; W is the n×c covariate matrix; a is the c × 1 coefficient vector; x is the n × 1 genotype vector; B is the SNP effect; u is the n × 1 random effect vector; ε is the n × 1 error vector; τ is residual variance; λ is variance ratio; K is the n × n kinship matrix; I is the identity matrix. MVN denotes multivariate normal distribution.

The kinship matrix K was calculated as follows:K = −∑xi − 1nxixi − 1 xi where xi is the genotype vector for SNP i, xi is the sample mean, and 1n is a vector of one. GEMMA tested alternative (H1: B ≠ 0) and null (H0: B = 0) hypotheses for each SNP. Manhattan and QQ plots were generated in R 4.3.2, with thresholds determined by GEC.

## 3. Results

### 3.1. Phenotypic Trait Measurements

Fifteen morphological parameters were measured in 116 adult workers of *A. cerana cerana*. The results showed that the proboscis length ranged from 3.76 mm to 4.94 mm; femur length from 1.42 mm to 2.75 mm; tibia length from 1.67 mm to 3.01 mm; tarsus length from 1.11 mm to 2.05 mm; tarsus width from 0.66 mm to 1.31 mm; tergite III & IV length from 1.49 mm to 2.81 mm; sternum 3 length from 1.33 mm to 2.44 mm; wax mirror length on sternite III from 0.54 mm to 1.06 mm; wax mirror slanted length on sternite III from 0.69 mm to 1.80 mm; wax mirror interval on sternite III from 0.13 mm to 0.83 mm; sternite VI length from 1.36 mm to 2.36 mm; sternite VI width from 1.66 mm to 2.95 mm; forewing length from 3.94 mm to 9.29 mm; forewing width from 1.48 mm to 3.13 mm, and cubital index from 0.88 to 3.80 (Table 2).

### 3.2. Genetic Evolution and Population Structure Analysis

Whole-genome resequencing was performed for 116 *A. cerana cerana* samples. After quality control, a total of 270.11 Gb of clean read data was obtained. The average sequencing quality value (Q30) across 12 geographic populations reached ≥91.12%, with GC content ranging between 32.54–35.02%. Alignment of whole-genome resequencing data from 116 colonies with the reference genome sequence showed an average sequencing depth of 8.96×, a mean genome coverage of 97.68%, and an average mapping rate of 95.94%.

In an earlier study [33], we used the ADMIXTURE software (V1.3) to analyze population genetic structure and lineage information. Admixture results showed that when assuming that two subpopulations were present in the samples (K = 2), honeybee populations from the western highlands (NP, HS, SM, SL, XS, and ZS) were clearly differentiated from populations from eastern low hilly mountainous populations (WC1, WC2, ZA1, ZA2, CS1, and CS2) (Figure 1). Except for a few colony samples, 79.63% of samples contained at least two ancestral components, suggesting potential gene flow and relatively mixed genetic composition among different *A. cerana cerana* populations. The populations from the NP and HS regions were separated when setting K = 3 and K = 4. The cross-validation error rate was minimized when the K value was 2, indicating that this was the optimal K value. A PCA result supported these findings, with 116 samples from different locations distinctly clustered into two groups: high-altitude populations showed relatively dispersed distribution, while non-high-altitude samples formed tighter clusters (Figure 2). Tracy-Widom tests indicated the first three principal components explained 3.33%, 1.53%, and 1.52% of total genetic variance, respectively. Phylogenetic tree analysis further validated results from both population structure and PCA (Figure 3). These findings indicate that *A. cerana cerana* populations in the western plateau and the eastern hilly mountainous areas of Guizhou belong to two independent genetic groups, respectively.

### 3.3. Linkage Disequilibrium Analysis

Currently, biparental population linkage analysis and natural population association analysis represent two predominant methods for dissecting quantitative traits. Linkage disequilibrium (LD) refers to the non-random association of alleles at different loci during biological evolution, which may manifest as non-random combinations between: (1) two genes, (2) two markers, (3) two quantitative trait loci (QTLs), (4) a marker and QTL, or (5) a marker and gene [37]. The resolution of GWAS varies depending on LD magnitude and marker density. Using the filtered SNP matrix, we performed genome-wide LD analysis. Figure 4 illustrates the physical distances corresponding to LD decay. The observed patterns of intra-chromosomal LD among SNP loci in this natural population enable reliable GWAS implementation for trait dissection in Guizhou’s *A. cerana cerana* populations.

### 3.4. Genome-Wide Association Studies

Through alignment with the reference genome, a total of 27,361,052 single-nucleotide polymorphisms (SNPs) were obtained. After quality control filtering, 30,079 SNP loci were retained. These SNPs were then combined with 15 traits of the *A. cerana cerana*. A mixed linear model was employed to perform GWAS for each of these traits. Based on previous reports, the significance threshold for SNP detection was set at *p*-value < 10^−6^, adjusted by Bonferroni correction. The GWAS identified a total of 16 significant SNPs (Appendix A). No SNPs reached the significance threshold for proboscis length, femur length, tibia length, tarsus length, tarsus width, sternite III length, wax mirror slanted length on sternite III, sternite VI length, sternite VI width, forewing length, forewing width, cubital index, respectively (Figure 5A–E,G,I,K–O). For the tergite III & IV length, 12 significant SNP loci were identified (Figure 5F). Two significant SNP loci were detected for wax mirror length on sternite III (Figure 5H), and two additional significant SNP loci were found for wax mirror interval on sternite III (Figure 5J).

### 3.5. Candidate Gene Screening

To identify the gene regions containing the significantly associated SNPs and understand their potential pathways affecting major production traits in *A. cerana cerana*, we conducted locus information analysis, gene annotation, and functional analysis. By querying the NCBI gene database and examining 50 kb flanking regions of significant SNPs, we annotated these 16 SNPs (Appendix A). We identified 11 candidate genes significantly associated with tergum 3 + 4 length: *APICC_05611*, *APICC_06479*, *APICC_06480*, *APICC_02050*, *APICC_05706*, *APICC_02236*, *APICC_05419*, *APICC_07473*, *APICC_07474*, *APICC_09312*, and *APICC_09560*, annotated as involved Longitudinals, Zinc finger, Lamin, Leishmanolysin, Radial, Hypothetical, BTB/POZ, ATP-dependent, Nurim, and Dynein. For the wax mirror length on sternum 3, two significant SNPs were associated with *APICC_03144* and *APICC_08157*, annotated as Protein and hypothetical, respectively. For wax mirror spacing on sternum 3, seven candidate genes were identified: *APICC_00038*, *APICC_00039*, *APICC_06559*, *APICC_06952*, *APICC_08164*, *APICC_08492*, and *APICC_09369*, encoding Protein, Hypothetical, Cytosolic and Phospholipase proteins (Appendix A).

## 4. Discussion

In eusocial honeybees, the morphological differentiation of worker terga and sterna is not only crucial for individual development but also closely associated with labor behaviors such as honey storage and nest construction. This study represents the first GWAS identification in Guizhou *A. cerana cerana* of 12 SNPs associated with tergite III & IV length, 2 SNPs with wax mirror length on sternite III, and 2 SNPs with wax mirror interval on sternite III. These loci collectively annotate 20 candidate genes, providing novel insights into the molecular regulation of regional developmental divergence in Guizhou populations.

Insect terga and sterna are chitinous composites of cuticular proteins secreted by epidermal cells undergoing coordinated mitosis, differentiation, and morphogenesis during larval and adult stages [38]. Previous studies in *Bombyx mori* identified Notch and Hippo pathways as core regulators of tergum development [39,40], while *Drosophila HOX* genes mediate tergum—sternum patterning via Wnt/Dpp signaling [41,42,43]. Research on honeybees has revealed that in *A. mellifera*, caste-specific H3K4me1-marked promoters may drive worker developmental plasticity [44], potentially underlying regional morphological variation. Our study shows a multi-tiered regulatory network through 20 candidate genes. The annotation results suggest that these genes are functionally involved in BTB/POZ domains, phospholipase activity, and dynein-related processes. It is hypothesized that they may drive longitudinal tergal extension via conserved Wnt/β-catenin signaling, orchestrate myofibril or cytoskeletal alignment to affect tergal elasticity, direct organelle transport during wax mirror morphogenesis, or potentially mediate epigenetic regulation of developmental genes.

During insect development, wax gland development reflects gene-environment interplay, as evidenced by Himalayan *A. cerana* populations showing tergal/sternal dimorphism [45]. Other research shows that the intronic wax-mirror-associated SNP KZ288192.1:706123 may modulate chromatin accessibility via H3K4me1 remodeling [44], complementing the transposon-derived miRNA ame-mir-3721-3p (targeting DNMT3 in caste determination) in social insect developmental networks [46]. Environmental synergism is evident—nectar diversity drives sternal differentiation via Vg/JH pathway methylation [47]. The high coefficient variation (51.0%) in wax mirror spacing underscores strong selection pressure, with *APICC_09369* (phospholipase) potentially coordinating with *A. mellifera* Hex 70a wax synthesis pathways [48,49,50]. The discovery of this study suggests gap junction-mediated basal membrane tension as a novel plasticity mechanism.

Regional vegetation, altitude, climate, and floral resources collectively shape bee morphology [51,52,53,54,55] and even temperament [14]. Bees in florally diverse environments develop distinct wing dimensions, body mass, and foraging patterns compared to pollen-limited regions [53]. Nutritional stress differentially impacts colony phenotypes [56], particularly in foraging range and efficiency [57]. Monofloral diets (hazel, rapeseed, pine, etc.) alter tergal/sternal enzyme activities in *A. mellifera* [58], likely via amino acid-mediated developmental modulation [59]. Additionally, diet-induced gut microbiome shifts [60,61] may further amplify gene expression divergence underlying regional morphotypes. In this research, high-altitude populations (e.g., HS, XS) exhibited significantly elongated terga, with the GC-rich zinc finger gene potentially enhancing chitin synthase activity under hypoxia—contrasting with *A. mellifera*’s AT-rich gene adaptation [51], suggesting divergent evolutionary strategies. The high-altitude-enriched SNP implies directional selection on cytoskeletal polarity and nuclear stability to optimize honey storage.

It is noteworthy that the functional characterization of these candidate genes through RNAi and multi-omics approaches remains essential, given the limited annotation of *A. cerana cerana* genomes.

## 5. Conclusions

This study successfully identified SNPs and candidate genes associated with key morphological traits in the *A. cerana cerana* through GWAS. The discovered SNPs and their corresponding genes may regulate the lateral and longitudinal development of the tergum and sternum in the *A. cerana cerana*. As a pioneering investigation into key traits of the native *A. cerana cerana*, this research holds significant implications for improving honey crop capacity, foraging efficiency, and ultimately enhancing the economic benefits for local beekeepers. Future studies can integrate transcriptomic, metabolomic, and proteomic approaches to further elucidate how the screened SNPs influence phenotypic traits via regulating gene expression or metabolic pathways. Such efforts will help clarify their functional roles in morphological development and provide a theoretical foundation for conserving the genetic integrity and germplasm resources of the native Chinese honeybee.

## Figures and Tables

**Figure 1 genes-16-01148-f001:**
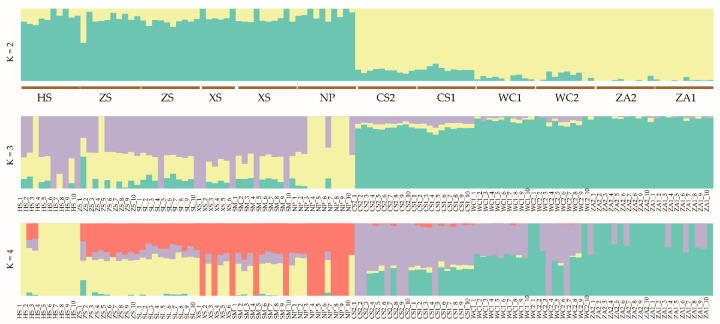
Analysis of the genetic structure of *Apis cerana cerana* populations. Note: ADMIXTURE analysis with K from 2 to 4. The *x*-axis represents each sample, and colors represent proportions of samples in each of the K inferred clusters.

**Figure 2 genes-16-01148-f002:**
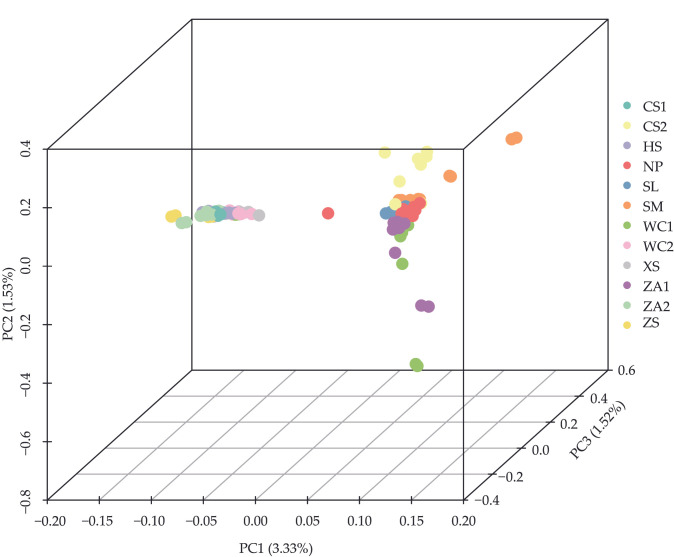
Principal components analysis of *Apis cerana cerana* in 12 areas of Guizhou Province. Note: The *x*-axis, *y*-axis and *z*-axis represent PC1, PC2 and PC3, respectively.

**Figure 3 genes-16-01148-f003:**
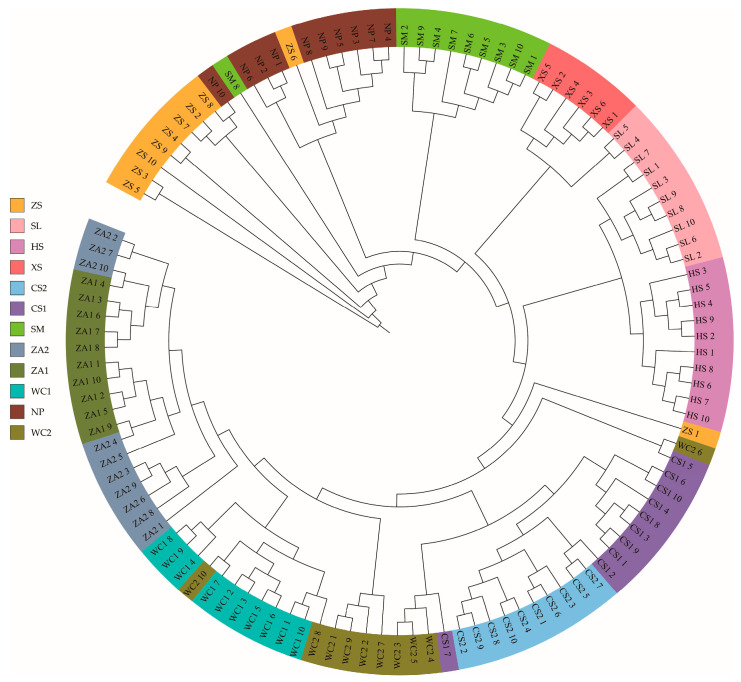
Phylogenetic tree of 116 *Apis cerana cerana* workers in Guizhou Province. Note: NP: Niupeng; ZS: Zhongshui; HS: Heishi; XS: Xueshan; SM: Shimen; SL: Shilong; CS1: Chishui 1; CS2: Chishui 2; ZA1: Zheng’an 1; ZA2: Zheng’an 2; WC1: Wuchuan 1; WC2: Wuchuan 2.

**Figure 4 genes-16-01148-f004:**
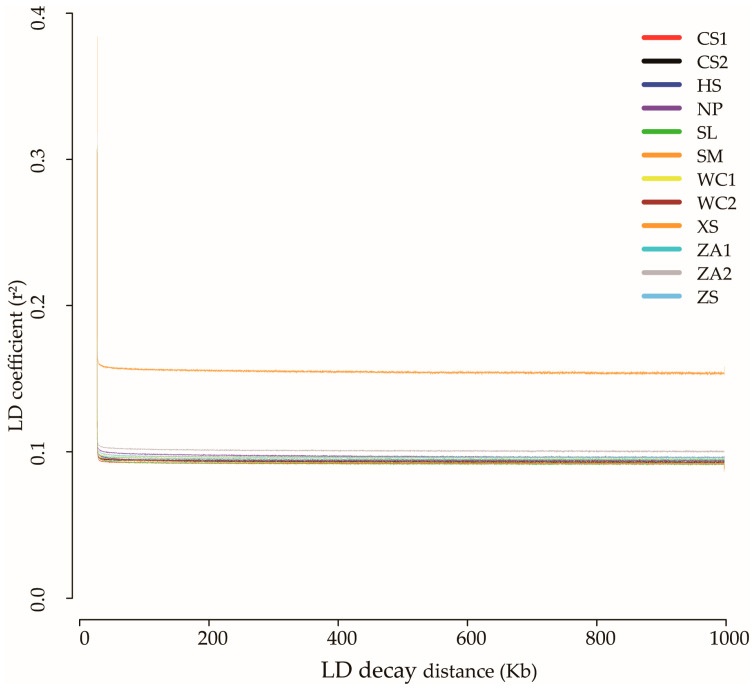
Linkage disequilibrium analysis of *Apis cerana cerana* in 12 areas of Guizhou Province. Note: NP, Niupeng; ZS, Zhongshui; HS, Heishi; XS, Xueshan; SM, Shimen; SL, Shilong; CS1, Chishui 1; CS2, Chishui 2; ZA1, Zheng’an 1; ZA2, Zheng’an 2; WC1, Wuchuan 1; WC2, Wuchuan 2.

**Figure 5 genes-16-01148-f005:**
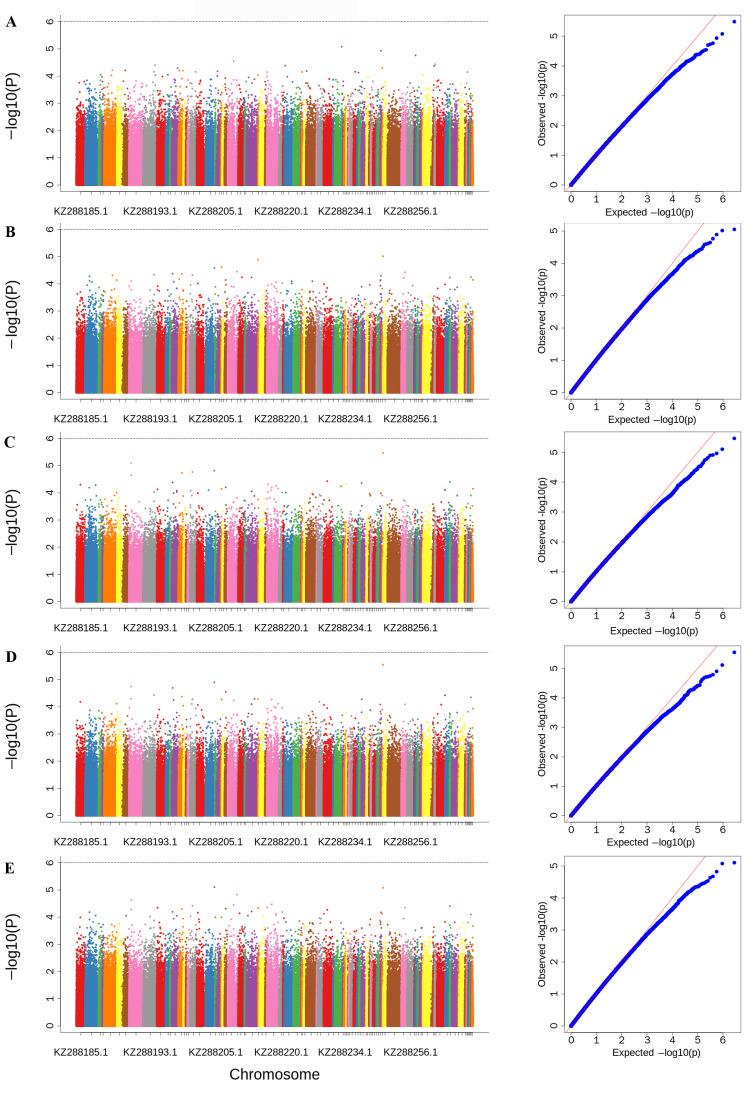
Manhattan (**left**) and QQ (**right**) plots of GWAS results for 15 traits in *Apis cerana cerana*. Note: The left panel shows a Manhattan plot, with the *x*-axis representing chromosomes and the *y*-axis indicating the value of −log10(*p*). The right panel displays a QQ plot, where the *x*-axis indicates the expected value of −log1o(*p*) (blue dots) and the *y*-axis indicates the observed value of −log10(*p*) (orange line). (**A**) Proboscis length; (**B**) Femur length; (**C**) Tibia length; (**D**) Tarsus length; (**E**) Tarsus width; (**F**) Tergite III and IV length; (**G**) Sternite III length; (**H**) Wax mirror length on sternite III; (**I**) Slanted wax mirror length on sternite III; (**J**) Wax mirror spacing on sternite III; (**K**) Sternite VI length; (**L**) Sternite VI width; (**M**) Forewing length; (**N**) Forewing width; (**O**) Cubital index.

**Table 1 genes-16-01148-t001:** Bee sample information.

Abbreviation	Sampling Site and Workers’ Repeated Number	Rearing Method	Longitude (E)	Latitude (N)	Altitude (m)
Niupeng (NP)	Fahong Village (10)	Movable-frame	103.77	27.12	1816.40
Zhongshui (ZS)	Huahongyuan Village (10)	Movable-frame	103.85	27.22	2071.00
Chishui 1 (CS1)	Malu Village (10)	Movable-frame	105.47	28.18	866.50
Chishui 2 (CS1)	Hongxin Village (10)	Movable-frame	106.18	28.51	549.80
Wuchuan 1 (WC1)	Huangyang Village (10)	Movable-frame	108.02	28.66	929.00
Wuchuan 2 (WC2)	Tongxin Village (10)	Movable-frame	107.93	28.73	745.60
Shimen (SM)	Quanfa Village (10)	Movable-frame	104.80	27.00	2247.90
Shilong (SL)	Shilong Village (10)	Movable-frame	103.93	27.40	1964.80
Zheng’an 1 (ZA1)	Jianshan Village (10)	Movable-frame	107.45	28.49	655.00
Zheng’an 2 (ZA2)	Miaoding Village (10)	Movable-frame	107.34	28.41	1138.40
Heishi (HS)	Shuiping Village (10)	Movable-frame	104.08	26.79	2482.30
Xueshan (XS)	Baimo Village (6)	Movable-frame	104.08	27.20	2370.70

**Table 2 genes-16-01148-t002:** Results of the *Apis cerana cerana* trait determination.

Traits	Mean	Standard Deviation	Max	Min	CV (%)
PL (mm)	4.46	0.27	4.94	3.76	0.06
FL (mm)	1.99	0.44	2.75	1.42	0.22
TL (mm)	2.31	0.50	3.01	1.67	0.22
TaL (mm)	1.54	0.35	2.05	1.11	0.23
TaW (mm)	0.96	0.21	1.31	0.66	0.22
T3&4L (mm)	2.46	0.14	2.81	1.49	0.06
S3L (mm)	1.89	0.37	2.44	1.33	0.20
WML3 (mm)	0.76	0.14	1.06	0.54	0.18
WMSL3 (mm)	1.28	0.26	1.80	0.69	0.20
WMI3 (mm)	0.31	0.16	0.83	0.13	0.51
S6L (mm)	1.85	0.38	2.36	1.37	0.21
S6W (mm)	2.29	0.47	2.95	1.66	0.21
FWL (mm)	6.91	1.58	9.29	3.94	0.23
FWW (mm)	2.45	0.50	3.13	1.48	0.21
CI	2.38	0.53	3.80	0.88	0.22

Notes: PL: proboscis length; FL: femur length; TL: tibia length; TaL: tarsus length; TaW: tarsus width; T3&4L: tergite III&IV length; S3L: sternite III length; WML3: wax mirror length on sternite III; WMSL3: wax mirror slanted length on sternite III; WMI3: wax mirror interval on sternite III; S6L: sternite VI length; S6W: sternite VI width; FWL: forewing length; FWW: forewing width; CI: cubital index.

## Data Availability

The research data can be review at https://www.scidb.cn/s/j6ZR7r (accessed on 21 August 2025).

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
