# Peer review of "Genome-Wide Association Studies of Key Traits in Apis cerana cerana (Hymenoptera: Apidae) from Guizhou Province"

_genes, 2025, doi:10.3390/genes16101148_

Round 1

Reviewer 1 Report

Comments and Suggestions for Authors

I have provided detailed comments directly in the PDF file for your reference. In general, the manuscript is scientifically sound, but I believe it can be further improved by:

  1. Figures and Tables: Please improve the resolution of figures, as some are difficult to read when printed. Captions should also be expanded with more explanatory details.

  2. Language and Style: Some sections need polishing for grammar, fluency, and logical flow.

  3. Discussion and Conclusion: These parts would benefit from clearer connections between results and interpretation, and a stronger emphasis on the novelty and implications of your findings.

Best regards

Comments on the Quality of English Language

The English could be improved to more clearly express the research.

Reviewer 2 Report

Comments and Suggestions for Authors

Dear Authors,

The manuscript requires some corrections and clarifications which are also found in the manuscript directly in the appropriate place. You can easily solve them. These are:

Line 13: Please elaborate on the abbreviation GVAS. Because it is the first mention in the manuscript.

Line 26: I suggest removing the keyword: Guizhou Province... because it is not common practice to mention the area in this section

Lines 30-31: Related to: <Honey bees are recognized as a kind of resource insects, due to their indispensable role in maintaining ecosystem balance and promoting sustainable development.>

Line 35: What do you mean by: the weather varies within 10 li (what does the word "li" mean?) Please write a universally valid unit.

Line 102: To Table 1: It is very important to make the connection with the text related to the table, because I did not find Table 1 in the textual description.

To Table 1 (content): What does the number (10) represent at each site? Could it be the repetition? What do you mean when you say size at the top of the table? The size of the site or the number of repetitions. Please clarify, because it is not clear.

To Figure 1: Since the characters of the values ​​are not readable, I suggest the authors provide the editor with the editable form and the data behind this mix of images. Either provide the individual images enlarged and clear. However, if you have already provided the editable form and the editor has concentrated the images into a single figure, then ignore the comment.

To Figure 2: The image is blurry and the colored lines are overlapping. Please redraw it or provide an editable form.

Line 230: To Table 4: Are you sure it's Table 4? Or is it Table 3? I didn't find Table 3 in the manuscript. Please clarify whether its absence is due to a context error or a typographical error.

If it is Table 3, then review the numbering of all tables and correct it both in the table heading and in the text.

To Figure 5: You should review Figure 5, section L because it has illegible characters on the x-axis and the diagram on the right has partially visible writing. Either provide the editable form to the editor who can intervene and arrange it correctly.

Line 263: Please write Table, not Tab.

Line 264: To Table 5:  Table 5 or Table 4? Clarify and write correctly and make the appropriate connection in the textual description.

Line 346: To References: Please review the References and write the journal titles correctly; sometimes you abbreviate, other times you do not, and sometimes you insert a dot between words, while other times you omit it.

Line 348: Please write Ecol. Evol.

Line 400: Please write Proc. Natl. Acad. Sci.

Line 380: Please write Nat. Rev. Methods Primers.

...and so on

Kind regards,

R

Reviewer 3 Report

Comments and Suggestions for Authors

Review

Title: Genome-Wide Association Studies of Key Traits in Apis cerana cerana (Hymenoptera: Apidae) from Guizhou Province

The paper aimed to identify genes linked to phenotypic traits in 12 the Apis cerana cerana through GWAS.

The paper is generally well written, but it has a few weaknesses that needs corrections. Please see my observations below.

Please use different keywords as in title. Please if possible do not use abbreviations in the Abstract, in its present form is hard to understand this part of the manuscript.

I especially appreciated the methodology, the presentations and the clear description of the works made.

However the results are clear, the figures must be improved. First, the Figure 1 description and legends must be expanded. Please add the methods here, and for A please improve the clarity. Also please explain for PCA analyses the factors you used for explaining axis x and y.

After figure 1 you have figure 4. With a short legend. Please correct here and explain where is 2 and 3. Also please consider that this figure 4 is not interpretable. Change or reformat.

Figure 5 is again strange, has short legends, and can not be interpreted. You have 3 times figure 5 presented as numbering, Please use a different format for this figure, please verify the numberings and please complete its explanations.

Again, Table 4 and 5 is not necessary here, omit or put in the supplementary file.

I wonder why after identified SNPs and candidate genes associated with key morphological traits in the A. cerana cerana are not correlated with environmental factors. An IndVal methods would explain better these variations. Othervise you have some genetic data related to the morphology, but no clear environmental or other trait associated explanations can be made.

Please reconsider to have this analyses included.

Please also add more detailed explanations to the Occlusions about the associations between these morphological variations and environmental traits.

Comments on the Quality of English Language

Review

Title: Genome-Wide Association Studies of Key Traits in Apis cerana cerana (Hymenoptera: Apidae) from Guizhou Province

The paper aimed to identify genes linked to phenotypic traits in 12 the Apis cerana cerana through GWAS.

The paper is generally well written, but it has a few weaknesses that needs corrections. Please see my observations below.

Please use different keywords as in title. Please if possible do not use abbreviations in the Abstract, in its present form is hard to understand this part of the manuscript.

I especially appreciated the methodology, the presentations and the clear description of the works made.

However the results are clear, the figures must be improved. First, the Figure 1 description and legends must be expanded. Please add the methods here, and for A please improve the clarity. Also please explain for PCA analyses the factors you used for explaining axis x and y.

After figure 1 you have figure 4. With a short legend. Please correct here and explain where is 2 and 3. Also please consider that this figure 4 is not interpretable. Change or reformat.

Figure 5 is again strange, has short legends, and can not be interpreted. You have 3 times figure 5 presented as numbering, Please use a different format for this figure, please verify the numberings and please complete its explanations.

Again, Table 4 and 5 is not necessary here, omit or put in the supplementary file.

I wonder why after identified SNPs and candidate genes associated with key morphological traits in the A. cerana cerana are not correlated with environmental factors. An IndVal methods would explain better these variations. Othervise you have some genetic data related to the morphology, but no clear environmental or other trait associated explanations can be made.

Please reconsider to have this analyses included.

Please also add more detailed explanations to the Occlusions about the associations between these morphological variations and environmental traits.

Author Response

Thank you very much for handling and reviewing our manuscript, and for providing us with very valuable feedback. We have made the changes and uploaded the new version. Please check again.

Thank you again and best regards.

Yours sincerely,

Yinchen Wang

List of responses:

1.Please use different keywords as in title. Please if possible do not use abbreviations in the Abstract, in its present form is hard to understand this part of the manuscript.

Response: The keywords have been appropriately modified. Abbreviations in the abstract have been given their full names.

2.I especially appreciated the methodology, the presentations and the clear description of the works made. However the results are clear, the figures must be improved. First, the Figure 1 description and legends must be expanded. Please add the methods here, and for A please improve the clarity. Also please explain for PCA analyses the factors you used for explaining axis x and y.

Response: We have added more detailed descriptions in the Results section. The figure legends have also been improved. Please review.

3.After figure 1 you have figure 4. With a short legend. Please correct here and explain where is 2 and 3. Also please consider that this figure 4 is not interpretable. Change or reformat.

Response: The previous version combined Pic2 and Pic3, but we forgot to update the numbering. This has been fixed in the new version.

4.Figure 5 is again strange, has short legends, and can not be interpreted. You have 3 times figure 5 presented as numbering, Please use a different format for this figure, please verify the numberings and please complete its explanations.

Response: We have improved the numbering and legends (Figure 5 consists of 3 panels, two of which are continuation figures). Please check in the new version.

6.Again, Table 4 and 5 is not necessary here, omit or put in the supplementary file.

Response: We have placed it in the supplementary file.

7.I wonder why after identified SNPs and candidate genes associated with key morphological traits in the A. cerana cerana are not correlated with environmental factors. An IndVal methods would explain better these variations. Othervise you have some genetic data related to the morphology, but no clear environmental or other trait associated explanations can be made. Please reconsider to have this analyses included. Please also add more detailed explanations to the Occlusions about the associations between these morphological variations and environmental traits.

Response: We appreciate you for raising this point. Since the primary purpose of GWAS is to identify SNPs associated with phenotypic traits, our study specifically focuses on detecting SNPs linked to the phenotypic characteristics of the Chinese honey bee (A. cerana cerana) through whole-genome sequencing data, and subsequently screening candidate genes based on these identified SNPs. Therefore, it is difficult to correlate the sequencing results from this study with environmental factors. However, our further research is progressing in that direction.

Round 2

Reviewer 1 Report

Comments and Suggestions for Authors

Dear Authors,
I have carefully reviewed the revised version of your manuscript. I appreciate the effort you have made to address the reviewers’ comments. The necessary edits and clarifications have been properly implemented, and the manuscript has significantly improved in terms of clarity and scientific presentation. I have no further concerns, and I believe the paper is now suitable for publication.

Reviewer 3 Report

Comments and Suggestions for Authors

Dear Editor and Authors
I am pleased to detect, that the authors made significant changes in their manuscript. My observations, except INDVAL statistics were followed, and answered, I also understand why this method has not been made. Therefore, I have no more comments.